# Sustainable Practices in Logistics Systems: An Overview of Companies in Brazil

**Vitor W. B. Martins** [1,2] , **Rosley Anholon** [1] , **Osvaldo L. G. Quelhas** [3] **and Walter Leal Filho** [4,*]

[1] School of Mechanical Engineering, University of Campinas—UNICAMP, Mendeleyev Street, 200, Campinas CEP-13083-860, Brazil

[2] Department of Production Engineering, State University of Pará—UEPA, Travessa Dr. Enéas Pinheiro, 2626, Belém CEP-66095-015, Brazil

[3] Laboratory of Technology, Business and Environment Management, Federal Fluminense University—UFF, Passo da Pátria Street, 156, Niterói CEP-24210-240, Brazil

[4] European School of Sustainability Science and Research, Hamburg University of Applied Sciences, Ulmenliet 20, D-21033 Hamburg, Germany

\* Correspondence: walter.leal2@haw-hamburg.de

**Abstract:** The main purpose of this article is to present an overview of the applications of sustainable practices in logistic operations performed by Brazilian companies. To reach this objective, the following steps were carried out: (1) a review of the literature on logistics systems and sustainability in logistics activities; (2) the collection of sustainability reports published by companies that perform logistics operations, which are recognized in Brazil; (3) a content analysis of the reports collected and (4) a discussion of the results, cross-checked with the literature and the extrapolation of conclusions. It was possible to identify 22 sustainable practices, and these practices were grouped into five macro areas. The authors of this paper believe that the findings presented here can be useful for professionals and researchers in the implementation of sustainability practices in logistics systems.

**Keywords:** sustainability; logistics systems; sustainability reports

## 1. Introduction

Since the publication of the 17 sustainable development goals proposed by the United Nations (UN) [1–5], sustainability has become an important aspect of business management. The definition of sustainability proposed by the World Commission on Environment and Development (WCEED) is the most accepted in both academia and business. According to this report, "Sustainable development is development that meets the needs of the present without compromising the ability of future generations to meet their own needs" [6], p. 41. Liu et al. [7] and Yun et al. [8] argue that this new context of business management requires organizations to critically evaluate their activities in relation to environmental, social and economic aspects.

Focusing on the industrial environment, Le et al. [9] argues that sustainability has become an important factor in competitive advantage for companies. Increasingly, society requires industries to develop sustainable practices throughout their manufacturing, supply and distribution operations [10,11]. Bask et al., Björklund and Forslund [12,13] corroborate this point of view which reinforces the need to better understand the role of sustainability in the development of logistics activities. Specifically, it can be seen in Brazil that the logistics sector has been evolving and consolidating rapidly in recent years to achieve a greater operational performance.

The inclusion of sustainable guidelines in the management of logistics activities is a topic of growing interest among researchers. Additionally, Bandeira et al. [14] and Stindt [15] argue that the

measurement of the sustainability of logistics activities can greatly contribute towards achieving the goals proposed by the UN. This is true specifically in relation to objective number 12, which is to ensure responsible production and consumption standards, as changes in production and distribution patterns are characterized as indispensable activities to reduce environmental impacts. However, according to Filho et al. [16], the requirement and importance of sustainable development goals are clear, yet there is a need for further research to clarify how these objectives can help to meet present and future sustainability challenges. According to Hong et al. [17], there is much to be researched in relation to the use of sustainability in logistics operations. These authors cite, for example, the need to better explore social aspects in these activities. Kim et al. [18] argues that there is still a lack of robust methods to analyze the social and environmental impacts of logistic activities in productive systems.

In recent years, sustainable logistics has been focused by academics, organizations and governments. This context can be defined as the analysis and promotion of sustainable procurement, sustainable transportation, sustainable packaging, sustainable distribution, reverse logistics, design and control of sustainable supply chain activities. It is important to highlight that most research carried out in this area focuses on the reduction of environmental impacts of logistics operations [19]. Therefore, it is important to mention that this research, besides analyzing sustainable practices in logistics operations related to environmental aspects, also presents discussions regarding social aspects.

Reefke and Lo [20] emphasize that the evaluation of overviews related to sustainable logistics can contribute to academic debates. Analyzing the literature, research presenting an overview of sustainable practices developed by Brazilian companies in logistic systems was absent. Considering this, the main objective of this research was defined: to create an overview of the sustainable practices developed in logistics systems by companies in Brazil. To reach this objective, 30 sustainability reports published by Brazilian companies were analyzed through content analysis.

In addition to the introduction section, this article has four more sections. Section 2 is dedicated to the presentation of theoretical reference in logistics systems and sustainability in logistics activities. Section 3 highlights the methodological procedures utilized, in order to enable replications. Section 4 presents the results of this study. Finally, Section 5 presents the conclusions and final considerations. The acknowledgments and references used are listed at the end.

## 2. Theoretical Background

This section is divided into two parts. The first part presents concepts, definitions and discusses the importance of logistics systems for business success. The second part presents the current research in sustainability in logistics activities.

### 2.1. Logistics Systems

Logistics refers to the managerial aspects pertaining to the acquisition, maintenance and transportation of materials, people and facilities. Corresponding to the process of planning, implementing and controlling the optimized flow of commodities, services and information from origin to destination, in order to meet customer demand [21], p. 540, [22,23]. According to Ballou [23], a logistics system is composed of several components which can be divided into key activities (management of the levels of services offered to customers, transportation, inventory and information flow management and order processing) and support activities (storage, materials handling, purchasing management, packaging design and maintenance of information). Logistics activities are strategic for companies and are critical for business success. Logistic planning is integral for a business's competitiveness and, for this, it is necessary to improve the performance of operations, integrate sectors of the company and provide logistics service with quality [23]. Companies should also consider the impact of logistics activities in sustainability [24,25].

One of the main aspects of a logistics system to be considered is the selection and definition of transportation modal [26,27]. According to Hauger et al. [28], this strategy is critical to company success. Moreover, the correct choice of transport modes enables better environmental and economic

results [29]. Zhou et al. [30] also reinforces this argument, and states that this aspect is essential for a more green logistics system.

Another interesting element of logistics is the strategy behind the programming and routing of vehicles. This requires the analysis of cargo capacity and the definition and the number of routes to be calculated. The planning of these activities can lead to additional problems and costs [31–34]. The definition and pricing of freight is an important activity for logistics systems. There are many variables to consider in this pricing and, according to Gavriilidis et al. [35] and Sánchez-Díaz [36], sustainability aspects must be considered in this analysis.

It is also important to analyze the service level offered to the clients of the logistics system. Boyacı et al. [37] argue that the optimized use of resources in order to satisfactorily meet the demands should be considered as part of the level of service offered to customers. According to Melović et al. [38], the need to meet customer's demands has gained attention in logistic strategies of companies in recent years.

Efficient warehouse management can also provide interesting gains to the whole logistics system. It is fundamental to the achievement of strategic operations that the planning of operational configuration of a warehouse takes into consideration the strategies of operations [39–41].

Aspects related to the definition, evaluation and relationship with suppliers, as well as strategies to define the period and quantity of purchases, are a routine part of a logistics system. According to Karuna et al. [41] and Miranda et al. [42], the correct choice of suppliers substantially increases the chances of success and improvement of business competitiveness. This is corroborated by Shi and Fung [43] and Andreasen and Gammelgaard [44].

In many companies recently there has also been a focus on packaging projects. The correct structuring of packaging optimizes the handling of a commodity, enables multiple configurations of storage, reduces losses by faults and allows greater security, among others. Efficient packaging design ensures the reduction of operational logistics costs, generates greater productivity and ensures competitive advantages [45,46].

Considering the aforementioned, the complexity of managing a logistics system is considerable. However, well-directed efforts in this context can improve productivity and business competitiveness.

## 2.2. Sustainability in Logistics Systems

This section aims to present the most up to date research related to sustainability in logistics activities. Through the synthesis of the studies presented, it is possible to note the importance of social, environmental and economic guidelines in these activities.

Aiming to highlight the importance of sustainability in logistics systems for business competitiveness, Eroglu et al. [25] conducted a study to evaluate the reaction of the stock market to the achievements of sustainability awards in logistics by some companies. The findings indicated that the stock market reacts positively to the presentation of these types of awards. Additionally, it was identified that this positive reaction was more significant than the reaction to other similar situations. This demonstrates that even shareholders recognize the importance of sustainability as a strategic factor for companies' survival.

Lu et al. [47] analyzed the insertion of sustainability in logistics operations of container terminals. The effects of internal practices (related to communication) and the collaboration of external practices (regarding suppliers, customers and subcontractors) on sustainable performance of operations was analyzed empirically. The findings demonstrated that both internal practices and external collaboration positively affects the sustainable performance of operations. In addition, Lu et al. [48] analyzed the effects of sustainable management in logistics operations in the port context. The results reinforced the fact that external collaboration is positively associated with internal management, and that this positively influences the sustainable performance of port operations.

Rai et al. [49] proposed an extensive list of indicators relating to the transportation of cargo with an operational goal (in support of urban policies and planning) to improve the service of sustainability.

In the same theme, Andersson et al. [50] also developed a framework of indicators. However, the focus was to measure sustainable logistics innovation in retail operations.

Cherrafi et al. [51] analyzed the relationship between green innovation practices and the performance of logistics chains. The findings showed synergistic effects between the adoption of practices and performance. The study showed that the practices that most positively affected the performance of logistics chains were eco-design, life cycle assessment, green manufacturing, reverse logistics and waste management.

Morgan et al. [52] performed a study which looked at the factors that influence the success of implantation of sustainable practices in logistics chains. Among other factors, they highlighted the commitment of resources as a fundamental aspect in order to obtain good results.

Bask et al. [12] examined the function of environmental sustainability in transport operations of companies providing logistics services. The findings showed that carrier companies, which operate globally, are the most interested in environmental subjects. This is due in part to external pressures, and partly because they see transport sustainability as a potential source of competitive advantage. However, due to the lack of widely accepted methods to measure the environmental impact of transport, companies cannot easily share costs and benefits of initiatives taken among supply chain members, nor can they use initiatives as marketing arguments to differentiate their offers [12].

The study by Watanabe et al. [53] suggests that for good performance of logistics systems it is necessary to consider the efficient use of technological resources of transformation, information processing and operations of handling and transport. However, there are no standard criteria or rules to evaluate such activities in the context of sustainability, which makes it difficult for companies to understand.

Finally, it is worth highlighting other studies that have addressed sustainable logistics practices in different contexts. Firstly, Murphy and Poist [54] aimed, through an empirical study, to provide an overview of the social responsibility in logistics, identifying key factors, strategies and functional impacts. González-Benito and González-Benito [55] identified essential factors for the adoption of management practices in logistics that impact on the environmental performance of the company through the analysis of two variables (the environmental pressure of stakeholders and the beliefs of its managers). Kim and Lee [56] also considered in their study the pressure of interested parties; however, they analyzed the eco-oriented culture in the company. Additionally, Colicchia et al. [57] evaluated the adoption of environmental initiatives in the contract logistics sector. They identified facilities and difficulties in the adoption of such initiatives, presenting new insights in theory and opening opportunities for new research on these topics.

Analyzing the research mentioned above on sustainable logistics, it is possible to verify the variety of objectives they present. This shows the efforts in literature to enlarge the debate on this topic. It is possible to highlight these efforts in the following themes: proposal of indicators to evaluate logistics operations; evaluation of the sustainable logistics impacts in a company's performance; identification of the factors that influence sustainable practices in logistics systems; analysis of the environmental aspects in the transportation sector and lack of criteria to evaluate the sustainability in logistics operations. Only one study had as main objective to give a general overview of social aspects in logistics; however, it had a different focus from our research.

According to the research presented, there is no article presenting an overview about the insertion of sustainability in logistics operations in Brazil. This research aims to fill this gap.

## 3. Methodology

For the development of this research, the following steps were carried out: (1) review of the literature on logistics systems and sustainability in logistics activities, to create a theoretical basis; (2) collection of sustainability reports published by companies that perform logistic operations and that are recognized in Brazil; (3) content analysis of the reports collected following the recommendations presented by Elo and Kyngäs [58]; and (4) discussion of results with the literature and establishment of conclusions. Figure 1 summarizes the steps of research.

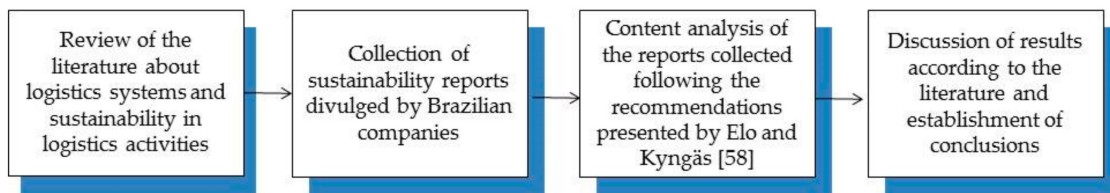

**Figure 1.** Steps of the research. (Source: Authors).

The literature review was conducted in the following scientific bases: Science Direct, Taylor and Francis, Springer, Emerald Insight and Wiley. Initially, to understand the concepts associated with logistics systems, the following terms were used: "logistics operations", "logistics systems", "warehouse management", "purchasing management" and "supplier management". Subsequently, in order to hone the search, the following terms were used: "sustainability in logistics systems" and "sustainability in logistics operations". Several articles were analyzed in detail. The synthesis of the articles most related to this research was presented in Section 2.

The collection of sustainability reports was done through companies' websites and published documents. There are some companies in Brazil that do not publish their sustainability reports annually, therefore, we considered the newest documents (between 2015 and 2018). In total, 30 reports were collected which comprise the sample considered in this study. According to the Global Reporting Initiative (GRI) [59], the reports enable organizations to publish their economic, environmental and social results. Additionally, they are able to present their strategies for contributing to sustainable development. Corporate sustainability reports have been used over the years by many researchers around the world, as they consolidate a large amount of relevant information regarding sustainability to be discussed and analyzed [60,61]. It should be noted that no studies were found that analyzed sustainable practices in logistics systems through information in sustainability reports published by companies.

Firstly, we did the sample characterization. After this, the reports were analyzed using the guidelines proposed by Elo and Kyngäs [58] in order to carry out the analysis of the content. According to these authors, a content analysis can be carried out in three phases: (a) preparation phase; (b) organizing phase; and (c) reporting the analyzing process and results.

In the preparation phase, the researchers must define the unit of analysis. Polit and Beck (2004), cited by Elo and Kyngäs [58], p. 109, argue that the unit of analysis may be a word or a theme. In our research, the unit of analysis was defined by the following theme: "Sustainable practices developed by Brazilian companies in logistics systems". In terms of sample size, Duncan (1989), cited by Elo and Kyngäs [58], p. 109, argue that it should be representative. For this article, we selected 30 companies that were representative of the Brazilian economy and that had published sustainability reports between 2015 and 2018. Still in the preparation phase, the authors of this study read the sustainability reports to become familiar with the subject. Polit and Beck (2004), cited by Elo and Kyngäs [58], p. 109, recommend this familiarization, and highlight that no detailed analysis should be done without the researchers becoming familiar with the information.

After the preparation phase, there is the data organization phase. Elo and Kyngäs [58] mention that in this phase the study may assume deductive or inductive characteristics. Deductive analysis is performed when researchers aim to evaluate data using theories and models. Inductive analysis is recommended when there are no previous studies about the phenomenon, or the knowledge is fragmented. As previously mentioned, the study of sustainable practices applied to logistics activities developed by Brazilian companies is original and the current studies are focused on specific points and cases. Thus, we understand that the inductive analysis proposed by Elo and Kyngäs [58] is more adequate for our analysis.

For inductive analysis, Elo and Kyngäs [58] recommends performing the following steps: open coding, category creation and abstraction. Firstly, in the open coding phase, all material is analyzed and categories are freely created. Burnard (1991), cited by Elo and Kyngäs [58], p. 111, and Cavanagh

(1997), cited by Elo and Kyngäs [58], p. 111, argue that category creation increases the understanding of the phenomenon studied. In this research, during the coding phase, all sustainability reports were read and categories related to the information mentioned were created. Examples of these categories are: "practices associated with storage", "practices associated with vehicle routing", among others.

Following the recommendations of McCain (1988) and Burnard (1991), cited by Elo and Kyngäs [58], p. 111, the next step performed was to group the categories in "higher order headings", known as "macro areas". This was performed because many of the generated categories presented similar information, or they were close to each other in terms of themes. For example, practices related to modal choice, vehicle use and routing are close to each other in terms of themes, therefore, they were grouped together. Dey (1993), cited by Elo and Kyngäs [58], p. 111, points out that grouping categories into "higher order headings" requires comparisons among the data collected and this was done in this research. As a result of the grouping process, it was possible to obtain five macro areas; namely: 1: practices related to modal choice, vehicle use and routing; 2: warehouse practices; 3: practices related to suppliers and purchasing processes; 4: practices related to packaging management; and 5: expansive management practices and social programs.

The third phase of the inductive analysis is "abstraction". According to Polit and Beck (2004), cited by Elo and Kyngäs [58], p. 111, in the abstraction phase the researcher tries to create a general formulation. In this research, during this phase, we aimed to establish an overview of the application of sustainable practices developed by Brazilian companies in logistic activities. According to Polit and Beck (2004), cited by Elo and Kyngäs [58], p. 112, when the authors show a link between data and results, the reliability of the study is increased; additionally, the authors must describe the results in detail and tables and attachments are helpful in this way to increase the reliability.

It is important to remember that Elo and Kyngäs [58] argue that content analysis is characterized by a great challenge and it must be very flexible. The same authors point out that there is no one correct way to perform the analysis of the content. Because of this, it is very important to describe all the steps taken and highlight the research limitations. The detailed description of this work in this section met this objective. We describe in detail the procedures performed, allowing other researchers to understand the steps done and increase the research reliability. The authors of this paper checked the results carefully and obtained a consensus that the findings are reliable.

The last stage of this research was characterized by the comparison between the results obtained in this research and information presented in the literature, the establishment of conclusions and writing of a final report. This phase corresponds to the stage "reporting the analyzing process and the results," mentioned by Elo and Kyngäs [58].

## 4. Findings and Associate Discussion

The sustainability reports that were analyzed were published by companies operating in Brazil. These companies belong to 13 different economic segments, with emphasis on transportation (23%), food and beverages (17%) and e-commerce (13%). Figure 2 shows more details. It is worth mentioning that most of the reports analyzed follow the structure proposed by the GRI (83% of the reports), which facilitated the analysis. The authors of this paper believe that the fact that most of the companies use the GRI standard provides greater credibility to the information presented.

Table 1 presents the sustainable practices developed by companies grouped into five macro areas. The procedure used to group the practices was described in Section 3. It is important to emphasize that the five macro areas listed are found in logistic system components presented by Ballou [23]. We did not identify sustainable practices in some components presented by the mentioned author, thus, some components present by Ballou [23] were not listed in Table 1. Macro area 5 contemplates expansive management, sustainable practices and social programs. Figure 3 presents the percentage of companies that developed each sustainable practice identified in the macro area logistics considered.

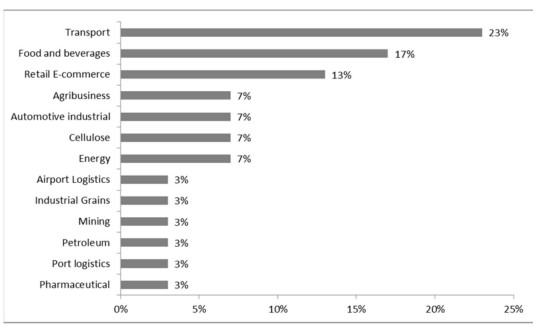

**Figure 2.** Distribution of companies analyzed according to segment of performance. (Source: Authors).

**Table 1.** Detail of the sustainable practices identified in each macro area. (Source: Authors).

| **Macro Area 1: Practices Related to Modal Choice, Vehicle Use and Routing** | |
|---|---|
| (1) | Use of electric vehicles. |
| (2) | Use of strategies to define the modal of transport to be used considering its transport capacity and the reduction of environmental impact. |
| (3) | Mapping the energy and/or fuel consumption of the vehicle used. |
| (4) | Planning of delivery routes contemplating the optimization and reduction of emissions of polluting gases. |
| **Macro Area 2: Warehouse Practices** | |
| (1) | Ergonomics and operational safety of the warehouse to improve working conditions. |
| (2) | Rainwater collection for use in warehouse operations. |
| (3) | Use of photovoltaic panels for power generation in warehouses. |
| (4) | Use of electric trolleys in warehouses. |
| (5) | Use of less polluting fuels in forklifts. |
| (6) | Mapping of waste in order picking operations. |
| **Macro Area 3: Practices Related to Suppliers and Purchasing Processes** | |
| (1) | Selection of suppliers considering their environmental practices, compliance with labor rules, code of conduct, anti-corruption program and sustainable certifications. |
| (2) | Programs for the development of suppliers with sustainable aspects. |
| (3) | Inclusion of social criteria and human rights in purchases processes. |
| (4) | Prioritization of purchase of inputs available in the local community. |
| **Macro Area 4: Practices Related to Packaging Management** | |
| (1) | Reuse, recycling and reverse logistics of packaging. |
| (2) | Reduction of the use of supplies in production of packaging. |
| (3) | Use of recycled material in the production of packaging. |
| (4) | Development of campaigns to encourage the use of more sustainable packaging. |
| (5) | Packaging design focused on the optimization of handling. |
| **Macro Area 5: Expansive Management Practices and Social Programs** | |
| (1) | Free transport for social projects. |
| (2) | Management actions focused on transparency and anticorruption. |
| (3) | Adoption of environmental and social rules such as ISO 14001 and Labor rules. |

Analyzing macro area 1 of logistics (practices related to modal choice, vehicle use and routing), it is possible to see that there are four different sustainable practices being developed by the analyzed companies. Among these, two stand out: (a) use of strategies to define the modal of transport to be used considering its capacity as transport and reduction of environmental impact, and (b) planning of delivery routes considering the optimization and reduction of emissions polluting gases. Both practices have been developed by only 27% of the companies analyzed. The other two practices identified in this macro area have an even lower percentage, with only 7% of companies using electric vehicles, and 10% of them performing mapping of the energy and/or fuel consumption of the vehicle used. Thus, based on the data reported, it is possible to see a small number of Brazilian companies

that have been developing sustainable practices related to modal choice, vehicle use and routing. This finding is worrying. According to Breunig et al. [31], scheduling and routing of vehicles can contribute significantly to a more sustainable development. This is corroborated by Hauger et al. [28], Zhou et al. [30] and Laghaei et al. [62].

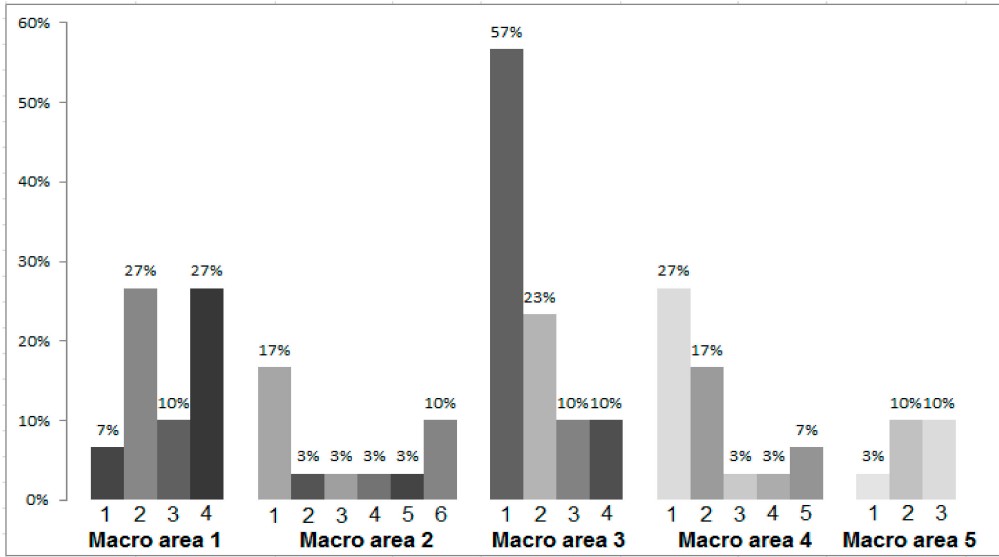

**Figure 3.** Percentage of companies that develop each practice identified in function of the logistic macro area (Source: Authors).

Considering macro area 2 (warehouse practices), it can be seen that it presents the greatest diversity in terms of sustainable practices identified (6 in total). Two of which should be highlighted: (a) ergonomics and operational safety of the warehouse to improve working conditions (with 17% of companies developing this practice) and (b) mapping of waste in picking operations (with 10% of companies developing this practice). The other practices are punctual, since the index of 3% corresponds to only 1 company. These practices are: rainwater collection for use in warehouse operations; use of photovoltaic panels for power generation; use of electric carts; and, use of less polluting fuels in forklifts. Similar to the previous macro area, it is noted that for the practices related to storage, the number of companies that have developed sustainable practices is still low. According to Makaci et al. [40], there is great potential for the development of sustainable practices in storage operations which can contribute towards a more sustainable future. Kang [63] argues that sustainable operations in warehouse activities improve productivity, safety and reduce adverse environmental impacts.

Macro area 3 (practices related to suppliers and purchasing processes) presents four different sustainable practices being developed by the analyzed companies. The practice regarding the selection of suppliers considering their environmental practices, compliance with labor rules, code of conduct, anti-corruption program and sustainable certifications stood out as the most developed practice by companies, with 57% occurrence. The practice of providing programs for the development of suppliers with sustainable elements, in turn, is developed by only 23% of the companies. The other practices in this macro area were identified in only 10% of the companies. According to Miranda et al. [42], the system of evaluation and careful selection of suppliers provides the observance of sustainable requirements and, in addition, positions the supplier as a strategic partner in the logistics operation. It can be seen, mainly through practice 1 of this macro area, that a good portion of the companies studied have been following the recommendations of Miranda et al. [42] and carefully selecting their suppliers. Despite the good results identified in practice 1, it is possible to observe many opportunities for improvement from a sustainable point of view. Large [64] argues that the achievement of sustainable goals requires joint actions between companies and their suppliers. Given the percentage reported in practice 2, it is still infrequent in Brazilian reality.

Considering macro area 4 (practices related to packaging management), it is possible to see that this area presents the second largest diversity of sustainable practices identified (5 in total). The most commonly used practices in this macro area are: (a) reuse, recycling and reverse logistics of packaging (being developed by 27% of companies) and (b) reduction of the use of inputs in the production of packaging (being developed by 17% of the companies). In a much smaller percentage of applications by the companies in the sample studied, there are practical packaging projects focused on the optimization of movement (7%); use of recycled material in the production of packaging (3%) and the development of campaigns to encourage the use of more sustainable packaging (3%). The authors of this article consider these numbers unsettling, as few companies develop activities that are integral for sustainable development. Crainic et al. [65] argues that packaging management allows a proactive integration of efficiency and attendance of sustainability aspects in logistics operations.

Analyzing macro area 5 (expansive management practices and social programs), it can be seen that this is the logistics area that presents the lowest diversity of sustainable practices (three in total). Practices regarding management actions aimed at transparency and anti-corruption, and the adoption of environmental and social norms are developed only by 10% of the companies. The practice of providing free transportation for social projects has an even smaller percentage, being developed by only 3% of the organizations. It can be seen that for the practices related to broad management systems and social programs, the number of Brazilian companies that are developing activities is still very low. Melović et al. [38] emphasize the importance of management systems for the continuous and sustainable growth of companies.

Finally, comparing the literature with the results obtained, we did not identify evidence of sustainable practices related to the following logistic activities: definition and pricing of freight; levels of services offered to customers and; execution of procedures for the operationalization of production and order separation. Therefore, it can be verified that there are many possibilities to develop and apply sustainable practices in these activities. The importance of points mentioned are corroborated by Sánchez-Díaz [36] and Lam et al. [66].

## 5. Conclusions

Based on the results presented, it is concluded that the main objective proposed for this research was achieved, as it was possible to develop an overview of sustainable practices performed by Brazilian companies in logistics operations. It was possible to identify a variety of sustainable practices developed within the five macro areas, but with a low degree of use by most of the companies analyzed. The practice that stood out was "supplier selection considering its environmental practices, compliance with labor standards, code of conduct, anti-corruption program and sustainable certifications". This practice is developed by 57% of companies of the sample.

It is important to emphasize the role of the logistic system in the pursuit of sustainable development goal number 12, proposed by the UN. The mentioned goal consists of ensuring responsible production and consumption, with a focus on supply chain operations; adequate logistics management can contribute to the improvement of every supply chain agent, from producer to final consumer.

Regarding the limitations of this study, it should be noted that the results and conclusions obtained were drawn by the information provided by 30 Brazilian companies in their sustainability reports. Other samples composed by different companies can provide other results and conclusions. Also, the analyses were performed using information from these reports, therefore the conclusions were based exclusively on them. Another limitation of this study is related to the content analysis. Remembering Elo and Kyngäs [58], this kind of analysis can be very flexible and the definition of the free categories and groups depends on the researchers.

Regarding the theoretical and practical contributions, the findings presented in this article have implications for theory and practice. The results provide an overview of the application of sustainable practices performed by companies in Brazil regarding their logistic activities, and highlight areas with improvement opportunities. Academics and professionals that work with logistic systems can use the

results presented here. Academics can use the results in classes or future research and professionals, in turn, can use the findings to enhance the sustainability in their companies.

Finally, we present some future research opportunities: (a) development of sustainable practices and tools to improve the insertion of sustainability in logistics activities; (b) developing roadmaps to help companies in Brazil with the transition to sustainable logistics; (c) proposal of models to assess the maturity of logistics systems in terms of sustainability; and (d) conducting sector studies, to present specific characteristics.

**Author Contributions:** All the authors contributed equally to this paper. They performed the collection of sustainability reports, analyzed the data, and wrote the paper together.

**Funding:** This research was funded by Conselho Nacional de Desenvolvimento Científico e Tecnológico—CNPq—Brazil, grant number 307536/2018-1 and 305442/2018-0. Universidade do Estado do Pará—UEPA—Brazil, grant number 626/18.

**Conflicts of Interest:** The authors declare no conflict of interest.

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
