# Peer review of "Sustainable Practices in Logistics Systems: An Overview of Companies in Brazil"

_sustainability, doi:10.3390/su11154140_

Round 1
Reviewer 1 Report
Please find the attached file for the review comments.

Author Response
Dear editor and reviewers,
Thank you very much for the considerations presented. These considerations have enabled a significant improvement in our manuscript. We have tried to accommodate all considerations, however, some of these suggestions differed between reviewers 1 and 2. As such, some points were improved by considering one point of view and the procedures performed by us in the research. We list below all considerations made and we justify the actions taken. Thank you for your much appreciated help.
Reviewer 2 Report
Title: Sustainable practices in logistics systems: an overview of companies in Brazil
Manuscript ID: sustainability-536651
First of all, thank you for the opportunity to review your paper. I understand that this paper aims to provide an overview of the sustainable logistics practices developed in companies in the context of Brazil. The authors have achieved this study purpose by conducting content analysis on 30 companies’ sustainability reports. I believe that this paper has the basics of a publishable paper; yet there are some issues in the current version of manuscript, which might need improvement.
Introduction
(1) It is not clear why Brazil? The authors should better justify why it is needed to analyze sustainable logistics practices in the context of Brazil.
(2) I feel your background of the study (2-3 paragraphs) and research aim (4 paragraph) do not match. Please better link them. Also, maybe you want to mention like despite the importance of sustainability, no studies have been done yet to provide an overview of sustainability practices for logistics operations?
(3) In addition, I think you should also need to spend some time justifying why the provision of the overview of sustainable logistics practices is important. In my view, the absence of this makes your purpose very weak.
Theoretical background
(3) In the first subsection (2.1), you mention logistics system is critical to company success (85 line on p. 2). Please explain more about this for nonprofessional readers, especially about ‘how’.
(4) In the next subsection (2.2), you just summarize current sustainable logistics-related studies one by one. In my view, this looks not a good literature review. Please synthesize those studies more, thereby finding out e.g., commonalities among them.
(5) I strongly recommend the authors to link their study to prior ones that deal also with sustainable logistics practices but in different contexts. Notable examples are: Murphy and Poist (2002) – USA, Gonzalez-Benito and Gonzalez-Benito (2006) – Spain, Kim and Lee (2012) – S. Korea, and Colicchia et al. (2013) – Italy. There are some benefits for doing this. First is to improve the literature review part of your study. Second is to link your study to another stream of sustainable logistics research (thus would attract more readers). Third is to enhance your originality, given that your study seems to be the first in the context of South America.
Methodological procedures
(6) On p. 5, you mention like there are a number of studies that examine corporate sustainability reports. If there are also some in sustainable logistics research, you should mention this, and differentiate your approach over them. Even if not, you may also want to mention this as it will improve your originality.
Findings and associated discussion
(7) All these 30 Brazilian companies are manufacturers? Or retailers as well? Please specify this.
(8) You are basically conducting content analysis. For this kind of analysis, I strongly suggest you to check inter/intra-rater reliability for the micro/macro items shown in Table 1. I won’t prescribe a specific test here, but by doing so this kind of test would surely be helpful for adding more rigor to your analysis. Your methodological rigor is quite weak in the current version of manuscript.
(9) I feel you now provide an overview of the current sustainable logistics practices in the context of Brazil. But this section looks too descriptive to really draw implications for research and practice. This might be because of the limited justification for your study; that is, as I said above, the reason why we need to know the overview? So please better address this ‘reason’ in the introduction part, and then interpret your results while answering it here in this section.
Conclusion
(10) You argue that a certain practice (macro 3-1) is quite necessary for all other companies in Brazil (line 288-289). On what basis do you make such an argument? In my view, ‘popularity’ (57%) does not necessarily represent a ‘necessity’. For this argument, I think you need to give more statements like macro 3-1 is a crucial way of achieving sustainability in logistics (evidence also needed). But raking each practice (Table 1) seems to be beyond the scope of the study?
Thus, I instead suggest you to keep your study aim (providing overview), and to better discuss study implications for research and practice. You might want to answer questions like what would be the role of your findings for the literature? what implications or guidelines could you give to practitioners? So please expand the conclusion part by providing more discussions for research and practice.
(11) Finally, the paper is well-structured, but there are many grammar glitches and awkward phrasings that should be corrected. I encourage the authors to use professional editing service to tighten-up the writing.
I hope you will find my comments useful. All the very best with your work.
References
Colicchia, C., Marchet, G., Melacini, M., Perotti, S., 2013. Building environmental sustainability: empirical evidence from Logistics Service Providers. J Clean Prod 59, 197-209.
González-Benito, J., González-Benito, Ó., 2006. The role of stakeholder pressure and managerial values in the implementation of environmental logistics practices. Int J Prod Res 44, 1353-1373.
Kim, S.T., Lee, S.Y., 2012. Stakeholder pressure and the adoption of environmental logistics practices Is eco-oriented culture a missing link? Int J Logist Manag 23, 238-258.
Murphy, P.R., Poist, R.F., 2002. Socially responsible logistics: An exploratory study. Transport J 41, 23-35.
Author Response

(The authors gave the same response as above.)

Round 2
Reviewer 1 Report
The authors had described more clearly about how their observations were obtained. The text was suitably modified and make it easier to follow their observations. This author recommends the paper sustainability-536651 for the publication in Journal of Sustainability.
Author Response
Rebuttal Letter
Manuscript ID: sustainability-536651
Title: Sustainable practices in logistics systems: an overview of companies in Brazil.
Dear editor and reviewer 2.
Thank you very much for the considerations presented in round two. These considerations have enabled a considerable improvement in our manuscript. We try to heed all considerations. We list below the considerations made and we justify the actions taken. Thank you for you precious help.

Reviewer 2 Report
I commend your efforts. Many thanks for addressing my comments. I believe that the revised manuscript is much stronger. Yet, the following very minor concerns remain:
Introduction
(1) It might be even much better if you could give a proper definition of sustainable logistics. By doing so, you may want to highlight your argument by saying that most of the studies focus on green issues but you do address social matters as well?
(2) I cannot agree with your statement “it is a topic few have explored in”. In fact, there are many. A simple google scholar search can provide a bunch of studies on this topic. But, again, maybe you can improve your study originality by highlighting that you focus on both green and social, as the extant literature focuses mostly on green issues. In that way, I think you can argue that “[sustainable logistics] is a topic few have explored in”.
Best of luck!
Author Response

(The authors gave the same response as above.)
